# High Mobility and Flexibility in the Habitat Use of Early Juvenile Pikeperch (*Sander lucioperca*) Based on a Mark-Recapture Experiment

Petr Blabolil [1,2,]*, Tomáš Jůza [1], Martin Čech [1] and Jiří Peterka [1]

1   Biology Centre of the Czech Academy of Sciences, Institute of Hydrobiology, Na Sádkách 7,
    370 05 České Budějovice, Czech Republic; tomas.juza@hbu.cas.cz (T.J.); martin.cech@hbu.cas.cz (M.Č.);
    jiri.peterka@hbu.cas.cz (J.P.)
2   Faculty of Science, University of South Bohemia, Branišovská 1760, 370 05 České Budějovice, Czech Republic
*   Correspondence: petr.blabolil@hbu.cas.cz; Tel.: +420-387-775-838

**Abstract:** Disentangling the role of factors responsible for juvenile fish dispersal is essential to understand the ecology of individual species, setting the corresponding conservation status and evaluating the potential risk in case of invasion. Because of their small body size and high sensitivity to environmental conditions, juvenile fish movements have largely been explained by external factors such as wind-induced water currents. In this study, early hatched pikeperch (*Sander lucioperca*) of hatchery origin were marked with oxytetracycline hydrochloride, stocked into a bay near the dam of a deep reservoir, and then monitored at approximately 10-day intervals using fix-frame trawling for 43 and 51 days after stocking, in 2007 and 2008, respectively. In both years, marked pikeperch were captured throughout the study period in the bay and closed dam section of the reservoir. After one month, individuals were captured in the middle section of the reservoir, approximately 5 km upstream from the stocking site. Four individuals were recaptured in the tributary section of the reservoir, about 10 km upstream from the stocking site during the last sampling in 2007. The farthest distance detection followed periods of strong wind. During daytime sampling, marked pikeperch were captured in both the warm epipelagic layer above the thermocline and the cold bathypelagic layer below the thermocline. The later sampling represented a community of vertically migrating individuals originally thought to consist only of reservoir-born and reservoir-experienced fish. This study suggested the high mobility and flexibility of 0+ pikeperch, as well as their unexpected behavioral plasticity.

**Keywords:** horizontal distribution; larvae; locality; trawling; vertical distribution

## 1. Introduction

The dispersal of fish in early life stages can have significant effects on the growth and survival of individuals and ultimately on population dynamics [1]. Dispersal in large freshwater systems is influenced by water currents, and previous studies have often considered juvenile fish to act similar to passive particles [2,3]. However, recent research suggests that juvenile fish dispersal may not be entirely passive [4]. The marine origin of percids [5] likely explains the existence of the early pelagic phase in their life cycle [3,6]. After hatching in the littoral, larvae migrate to the pelagic zone, where they spend some time and disperse in a water body [7]. Most early life history studies in Europe have focused primarily on the European perch *Perca fluviatilis* [8]. In inland lakes, the pelagic phase of European perch begins shortly after larvae hatch and lasts for several weeks [9]. Pikeperch *Sander lucioperca*, as a close relative, apparently have very similar distribution patterns, as the larvae of both species usually account for the majority of catches in the pelagic layers of deep canyon-shaped reservoirs [3,6,10]. Early juvenile pikeperch were recorded to have undertaken diurnal migration from near the surface habitats at nighttime to deep benthic





habitats at daytime in shallow, well-mixed reservoirs [11] or to inhospitable bathypelagic layers with cold water and low oxygen concentrations in the deep, stratified reservoirs at daytime [6,12]. Because annual fluctuations in year class strength are common in pikeperch populations [13,14], understanding the ecology of early life stages, including dispersal ability, is the first step in uncovering the factors affecting variable population recruitment.

Mark–recapture experiments with larvae and early juveniles can significantly contribute to the understanding of the habitat use and dispersal abilities of 0+ pikeperch in large, heterogeneous systems, and reveal the peculiarities of its early life stages' ecology. However, gathering sufficient numbers of wild-origin, viable fish larvae in specific period is usually difficult compared to fish production in hatcheries with stabile control conditions [15,16]. There, the parental stock is often limited to a few individuals kept under artificial (hatcheries) or semiartificial (ponds) conditions, and repeated artificial spawning can result in inbreeding and a reduction in genetic diversity [17]. Typical fish traits under artificial selection are fast growth, a large body size, high food conversion into body weight (fish are often fattier), a lower metabolic rate and swimming ability (fins can be reduced), early maturation, and high fecundity [18]. In natural conditions, fish of hatchery origin can have a different phenotype (e.g., weaker coloration, worse adaptation to changes in environmental conditions) and behavior (e.g., more aggressive behavior, worse prey hunting and lower antipredation ability) compared to fish of wild origin, thereby resulting in lower survival and fitness [19,20]. Fish of hatchery origin are usually stocked at high densities of similar size individuals, and therefore, the stocking of early life stages enables better adaptation to a new environment via selection and cohort differentiation [21]. The use of different habitats such as epipelagic and bathypelagic layers had thus far only been described for percid fishes of reservoir origin (cf. [6,10]). Moreover, a recent study demonstrated 0+ pikeperch diurnal vertical migration is size-dependent, as older, larger and more pigmented pikeperch migrate to the bathypelagic layer compared to younger and smaller pikeperch, who stay in the epipelagic layer during the day [22].

Water stratification, zonation and flowing are important for fish distribution in heterogeneous water bodies such as reservoirs [23]. The water circulation within a reservoir can be affected by natural gravity (passing river/rivers), wind, temperature-dependent water density, artificial mixing by aerators, boat traffic, or water discharge for turbines. Deep valley reservoirs formed by the impoundment of a river are characterized by elongated morphometry with meanders and longitudinal physical and chemical gradients [24]. In addition to horizontal spatial heterogeneity, the pelagic environment also varies vertically, especially in reservoirs with a relatively long water retention time, which allow the development of thermal stratification with a sharp separation between the warm and oxygen-rich epipelagic layer and the cold, deeper layer with a low oxygen concentration [25]. The separation of horizontal layers during the season causes the plunging of the river water under the epipelagic layer; therefore, the surface water currents may be wind-driven [24].

The aim of this study was to evaluate the spatial dispersal of 0+ pikeperch in the thermally stratified, deep-valley Římov Reservoir as part of a mark–recapture experiment by marking of pikeperch larvae released in a semi-enclosed bay and then monitoring them in the reservoir during the first weeks after stocking using fixed-frame trawling. We hypothesize that marked pikeperch larvae and early juveniles (1) do not make significant migrations and are captured in or near the bay into which they were released, (2) because they do not originate from a reservoir, they do not make the vertical migrations described in wild populations and exclusively use the surface water layer during the day and (3) detection of marked pikeperch far from the stocking place will happen after windy periods, thus creating surface water currents.

## 2. Materials and Methods

### 2.1. Study Site

The study was conducted in the deep-valley Římov Reservoir (48.849 N, 14.489 E), the Czech Republic (Figure 1). The reservoir has a maximum depth of 45 m, an average

depth of 16 m, an area of 210 ha, and a volume of $33.1 \times 10^6$ m³. The Malše River is the main tributary with a long-term average annual water flow of 2.6 to 10.8 m³ s⁻¹ and an average theoretical water residence time of 92 days [26]. The surrounding landscape of Římov Reservoir, mostly represented by the former, deep river valley, is covered by full-grown forest (98% coverage), which together with meanders makes this water body relatively less vulnerable to winds and the resulting, damaging wave action (winds of a speed > 6 m s⁻¹ rarely occur [27]). Strahovská Bay, in which the fish were stocked (the largest on the reservoir at 6.7 ha), is located near the dam (Figure 1). The bay has an elongated morphology. The inflowing Velešínský Brook has a long-term average annual flow of 0.007 to 0.072 m³ s⁻¹ in the shallowest section. Its deepest point (27 m) is at its mouth, feeding into the lacustrine section of the reservoir.

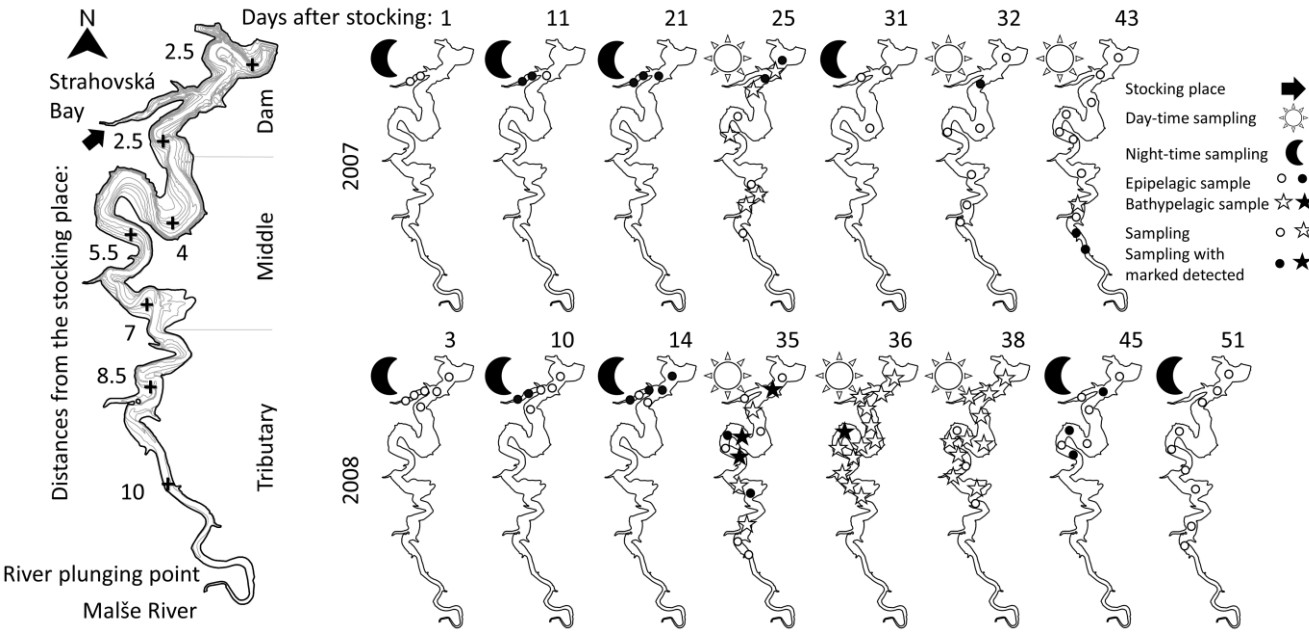

**Figure 1.** Map of the Římov Reservoir divided into reservoir sections, showing the distances from the stocking place in kilometers and isobaths in 5 m intervals (left side). Scheme of temporal and spatial sampling in the reservoir during the day and night in 2007 and 2008 (right side). Detection of marked pikeperch in specific places and depths is highlighted. Plots are North oriented. For details see legend and Table 1.

A longitudinal gradient of productivity is well developed in the reservoir, with the total phosphorus and chlorophyll-*a* concentrations decreasing from the tributary to the lacustrine section [28]. The reservoir is dimictic, with pronounced thermal stratification in summer (from mid-April to mid-October). The main purpose is drinking water storage; therefore, a long-term biomanipulative management strategy is implemented, aiming for a high abundance of piscivorous fish controlling planktivorous fish, and a high abundance of large Cladocera improving the water quality through filtration of phytoplankton. A viable pikeperch population has been established in the reservoir, but fluctuating natural recruitment has been observed from year to year [13,14]. Therefore, pikeperch have been stocked since 1979 at a rate of 2500 to 25,000 fingerlings per a year (early autumn fingerlings [26,29]).

**Table 1.** Details of fix-frame trawling, and pikeperch fry catch (numbers, No and mean standard length, SL ± standard deviation, SD) in 2007 and 2008 in Římov Reservoir. The day period represents 11:00–16:00 and night 22:00–3:00. The specific sampling locations are plotted in Figure 1.

| Year | Date | Day after Stocking | Sampling Period | Layer | Diameter of Trawl Opening (m) | Number of Hauls | Filtered Volume (m³) | Local Pikeperch | | Recapture Pikeperch | | Not Identified Captured Pikeperch | |
|---|---|---|---|---|---|---|---|---|---|---|---|---|---|
| | | | | | | | | No | SL ± SD (mm) | No | SL ± SD (mm) | No | SL ± SD (mm) |
| 2007 | 1.5. | 1 | night | epipelagic | 0.5 × 2 | 2 | 600 | 0 | | 0 | | 75 | 5.8 ± 0.4 |
| | 11.5. | 11 | night | epipelagic | 0.5 × 2 | 3 | 900 | 81 | 10.1 ± 1.2 | 14 | 8.5 ± 0.7 | 0 | |
| | 21.5. | 21 | night | epipelagic | 2 × 2 | 3 | 4460 | 188 | 14.3 ± 1.4 | 8 | 12.9 ± 0.6 | 890 | 12.8 ± 1.1 |
| | 25.5. | 25 | day | epipelagic | 4 × 2 | 5 | 12,840 | 122 | 14.0 ± 1.5 | 5 | 13.9 ± 0.8 | 73 | 13.0 ± 1.4 |
| | 25.5. | 25 | day | bathypelagic | 4 × 2 | 5 | 12,840 | 33 | 13.2 ± 1.5 | 0 | | 83 | 13.2 ± 1.5 |
| | 31.5. | 31 | night | epipelagic | 4 × 2 | 3 | 9800 | 31 | 13.9 ± 2.0 | 0 | | 2 | 16.5 ± 0.2 |
| | 1.6. | 32 | day | epipelagic | 4 × 2 | 7 | 14,016 | 28 | 14.6 ± 1.5 | 2 | 15.0 ± 0.7 | 12 | 13.8 ± 2.4 |
| | 12.6. | 43 | day | epipelagic | 4 × 2 | 10 | 35,040 | 113 | 14.9 ± 2.7 | 4 | 13.8 ± 1.7 | 322 | 12.3 ± 2.3 |
| | 12.6. | 43 | day | bathypelagic | 4 × 2 | 1 | 4200 | 6 | 17.5 ± 3.1 | 0 | | 10 | 16.0 ± 1.4 |
| 2008 | 24.4. | 3 | night | epipelagic | 0.5 × 2 | 6 | 3300 | 14 | 5.5 ± 0.2 | 0 | | 0 | |
| | 1.5. | 10 | night | epipelagic | 1 × 2 | 6 | 6820 | 36 | 6.2 ± 0.4 | 3 | 6.7 ± 0.3 | 0 | |
| | 5.5. | 14 | night | epipelagic | 1 × 2 | 6 | 6951 | 286 | 6.6 ± 0.6 | 26 | 8.0 ± 0.5 | 0 | |
| | 26.5. | 35 | day | epipelagic | 2 × 2 | 8 | 9404 | 290 | 11.2 ± 1.1 | 2 | 11.0 ± 1.4 | 461 | 10.2 ± 0.9 |
| | 26.5. | 35 | day | bathypelagic | 2 × 2 | 6 | 6048 | 92 | 12.3 ± 1.1 | 5 | 12.4 ± 0.7 | 401 | 11.5 ± 0.6 |
| | 27.5. | 36 | day | bathypelagic | 0.5/1/2 × 2 | 13 | 12,832 | 237 | 12.4 ± 0.9 | 1 | 13.0 | 1956 | 11.8 ± 1.3 |
| | 29.5. | 38 | day | epipelagic | 2 × 2 | 3 | 3660 | 106 | 12.4 ± 1.3 | 0 | | 421 | 10.1 ± 1.4 |
| | 29.5. | 38 | day | bathypelagic | 2 × 2 | 10 | 11,452 | 412 | 13.2 ± 1.2 | 0 | | 946 | 12.4 ± 1.2 |
| | 5.6. | 45 | night | epipelagic | 2 × 2 | 8 | 10,436 | 406 | 14.8 ± 1.8 | 4 | 13.9 ± 0.6 | 1429 | 12.9 ± 1.6 |
| | 11.6. | 51 | night | epipelagic | 2 × 2 | 9 | 8544 | 867 | 16.2 ± 2.3 | 0 | | 1047 | 13.5 ± 2.3 |
| Sum | | | | | | 114 | 174,143 | 3348 | | 74 | | 8128 | |

### 2.2. Fish Marking

Pikeperch of hatchery origin (1–3 days post-hatch) were marked with the antibiotic drug oxytetracycline hydrochloride (OTC) [30]. In 2007, 189,800 individuals and in 2008, 306,500 individuals at size (mean ± standard deviation, SD) 5.4 ± 0.3 mm SL were marked on 30 April and 21 April, respectively. In both years, larvae were treated in the Tisová hatchery (50.147 N, 12.606 E) through immersion in OTC at a concentration of 800 mg $L^{-1}$, a safe concentration proven for fixed mark deposited on calcified structures [30], transferred to the reservoir in nine 55 L barrels that were placed in water in the shallowest section of the Strahovská Bay (48.841 N, 14.471 E), and released after nine hours in the bath, when the water temperature had adjusted, during the nighttime (10–11 pm).

### 2.3. Fish Sampling and Processing

The samples of the targeted stocked fish were subsequently collected at approximately 10-day intervals (Table 1). The samples collected on 17 May 2008 were not frozen well enough, and not used for length or age data. Fixed-frame trawling with nets 2 m high and 0.5, 1, 2, and 4 m wide was used to collect fish (Table 1). The used trawl size was based on the 0+ pikeperch density to obtain a representative sample (tens of individuals from each haul). The mesh size was always 1 × 1.35 mm, with a collecting bucket at the end of the net [31]. The trawls were towed 100 m behind a boat at a speed of 1 m $s^{-1}$ for 5–10 min representing 250–580 m distance. The surface pelagic layer (0–2 m, hereafter referred to as the epipelagic layer) was sampled during both the day (11:00–16:00; i.e., full day) and night (22:00–3:00; i.e., full night) campaigns, and a deeper layer (7–13 m, hereafter referred to as the bathypelagic layer) was sampled during some daytime surveys to check the presence/absence of fish under the thermocline (Table 1, Figure 1). The vertical distribution of 0+ fish during the day (position of the bathypelagic layer in the water column) was determined using a SIMRAD EK60 scientific split-beam echo sounder operating at a frequency of 120 kHz. The transducer used (SIMRAD ES120-7C, vertical beam) had a circular beam pattern with a nominal angle of 7.1° (for details see [3]). The trawl path was measured using a GPS device (Garmin 60 CSx). A bathypelagic fish layer was never detected using hydroacoustics during the night, as due to the diel vertical migrations, fish were present in the epilimnion or in the littoral (cf. [6,10]), and therefore the layer was not sampled by trawl at night. The nets had weights attached to the lower part of the trawling frame, while a float was attached to the upper rim. For hauls in the bathypelagic layer, the length of the rope between the float and the frame corresponded to the depth of sampling [31].

After each haul, the catch was collected into a clean bucket. Within 20 min of being caught, 0+ pikeperch were manually selected. The selection was carried out by two or three trained researchers by spreading mixed random subsamples in a 1 cm layer in a white bowl. The pikeperch larvae are characterized by relatively large dark spots (chromatophors) randomly distributed on the sides of the body compared to other common species European perch (it has more small spots on the sides of the body in regular rows) [32], see Supplementary Material A. Pikeperch were selected by tweezers, placed in bottles uniquely labelled for each haul and frozen for laboratory processing. The other fish were anaesthetized and then preserved in a 4% formaldehyde solution. In the laboratory, within five months of capture, fish were thawed and once more verified for species identification under an Intraco Micro STM 8235410 N stereomicroscope (7–45 magnification) as pikeperch and European perch differ not only in their spots, but also by their numbers of post- and pre-anal myomeres [32]. Pikeperch were measured to standard length (SL) to an accuracy of 0.5 mm, and Sagittae otoliths for mark detection were extracted, cleaned from any attached tissues and glued to slides with thermoplastic adhesive (Crystalbond 509 clear). Otoliths were viewed using an Olympus AX70 microscope with fluorescent light (FITC filter set, extraction/emission wavelength 450–480/>515 nm, 200–600× magnification). When necessary for proper viewing, otoliths were sanded with 400- and 600-grit sandpaper [30]. Fish fixed in formaldehyde were identified to species, and 0+ pikeperch were counted and measured

as for the thawed sample. The other species in the catch was European perch (usually more abundant than pikeperch), and at the end of the sampling period, Cyprinids started to be caught (for the purpose of this study, species were not identified, but the most common are roach *Rutilus rutilus*, bleak *Alburnus alburnus* and freshwater bream *Abramis brama*) in the Římov Reservoir..

*2.4. Wind Measurements*

　　Wind velocity and direction data were recorded from a flouting meteorological tower (FIEDLER AMS s.r.o., České Budějovice, Czech Republic) installed 100 m from the nearest bank and 600 m from the mouth of the stocking bay in the dam section of the reservoir (48.849 N, 14.487 E). Wind data, for the period when the marked pikeperch were stocked and samples taken, were measured in 10 min intervals and graphically summarized using the windRose package [33] in R software [34].

**3. Results**

*3.1. Numbers of Pikeperch Captured in Different Years*

　　In 2007, during 39 hauls, 94,696 m$^3$ were filtered and 2102 0+ pikeperch captured. In the catch, 635 0+ pikeperch were examined in the laboratory, and 33 were identified as marked (i.e., 5.2%; Table 1). In 2008, the sampling effort was 75 hauls representing 79,447 m$^3$ of filtered water (compared to 2007, the tow durations had to be adjusted due to the presence of large phyto- and zoo-plankton and plugging of the net). In total, 9448 0+ pikeperch were captured, of these 2787 were examined and 41 identified as marked (i.e., 1.5%). Overall, only 74 larvae of the 496,300 larvae captured were positively identified as marked (0.015%).

*3.2. Dispersion of Marked Pikeperch along the Longitudinal Profile*

　　In both years, no marked pikeperch were captured during the first sampling campaign 1 and 3 days after stocking (Figure 1). Some 10 to 25 days after stocking, marked pikeperch were captured in the bay and nearby dam section (within 2.5 km from the stocking place) of the reservoir (Figure 1). Most of the marked pikeperch remained in the dam section of the reservoir during the whole study period, as marked pikeperch were captured here 32 days after stocking in 2007, and 35 and 45 days after stocking in 2008. Some individuals, however, migrated towards the tributary section, where four marked individuals were recaptured close to the tributary (approximately 10 km from the stocking place) in 2007, and eight marked pikeperch were captured in the middle section (within 7 km from the stocking place) of the reservoir in 2008 (Figure 1).

*3.3. Depth Distribution of Marked Pikeperch*

　　In 2007, the highest sampling effort was performed in the epipelagic layer, where all marked pikeperch were captured. In 2008, the 35th day after stocking, when epipelagic and bathypelagic layers were simultaneously sampled, marked pikeperch were captured in both layers. During the following campaigns, marked pikeperch were detected in the bathypelagic layer during the daytime (36th day after stocking), and in the epipelagic layer during the nighttime (45th day after stocking, when finishing the diel vertical migration; Table 1).

*3.4. Wind Measurements*

　　In both years, the wind direction was variable, with mild velocity prevailing in the dam section of the reservoir. The strongest wind intensities with a south-west direction (i.e., against the main natural water movement in the reservoir) were observed between 10 and 30 days after the release in 2007 and north-west direction between 21 and 30 days after the release in 2008 (Figure 2). Marked pikeperch were captured far from the release site 43 days after release in 2007, and 35–36 and 45 days after the release in 2008. Convenient directions of winds and the resulting waves might have pushed the surface water masses (and pelagic pikeperch) from the dam towards the tributary section of the reservoir.

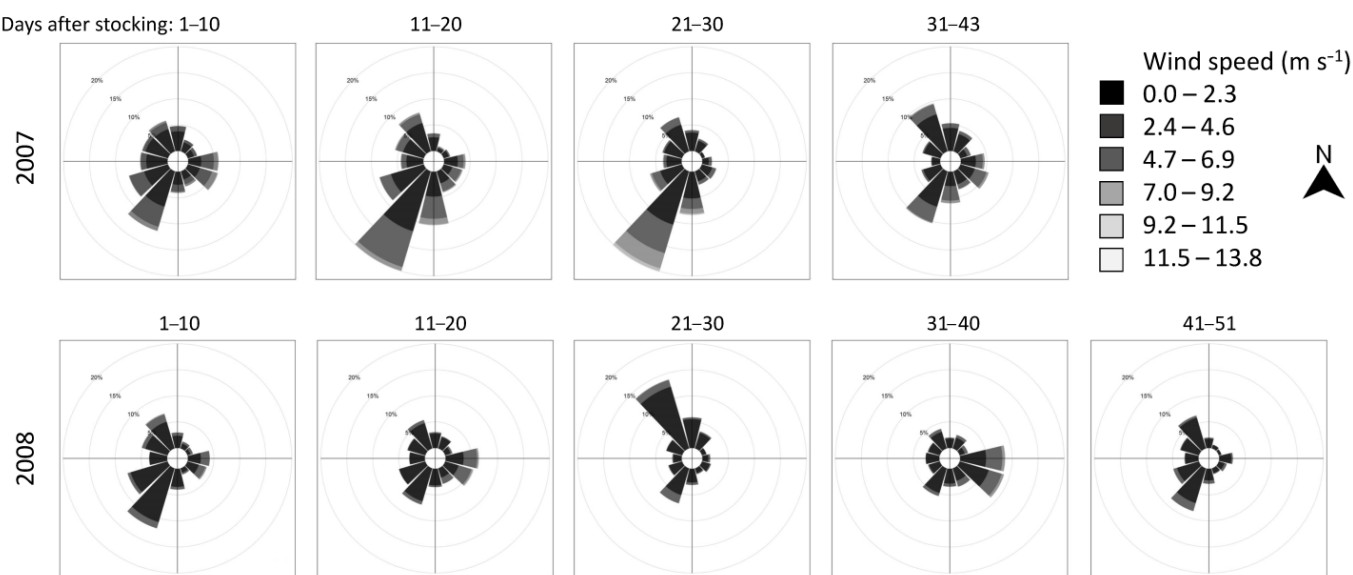

**Figure 2.** Wind rose diagrams showing wind velocity and vector frequency information recorded in the dam part of the Římov Reservoir during the marked recapture experiment in 2007 and 2008. Plots are North-oriented.

## 4. Discussion

Despite small body size and limited energy sources of 0+ pikeperch, some of the fish began to disperse spatially after spending some time in the bay in which they had been released. Surprisingly, usage of critically different pelagic habitats (epipelagic × bathypelagic) and diurnal vertical migrations of stocked 0+ pikeperch of hatchery origin were also observed; such behavior had previously only been attributed to percid fishes of reservoir origin (cf. [6,10]). The water currents probably have a significant impact on larval dispersal, even in a deep-valley reservoir. Some individuals showed site fidelity where they had been released, or migrated only slightly, while others were detected in the order of several kilometers along the longitudinal profile of the reservoir (upstream of the reservoir). This suggests that the movement activities of even stocked naïve pikeperch might be specific to the individual, highly variable, and dependent upon wind.

To better understand dispersal capabilities, a bay near the dam was selected as a simulation of the whole system of a large water body, where pikeperch spawn in bays and tributary sections ([35]; in the shallow part of the Strahovská Bay, the male pikeperch guarding the nests were observed by SCUBA divers in May 2011, [27]). Early hatched larvae inhabit shallow areas before filling the gas bladder [36]. This likely explains our failure to detect marked pikeperch within the first sampling campaigns, as trawling was conducted only in pelagic zone. In the following development phase, pikeperch is positively phototactic; after switching to exogenous feeding, pikeperch move actively to open water or the littoral, where their distribution is related to water currents [7]. The dam section of the reservoir is characterized as the most oligotrophic, with the lowest nutrient and consequently zooplankton density [37]. In general, because pikeperch is a species that prefers eutrophic to hypertrophic lakes characterized by shallow mean water depth and low water transparency [38], their larvae are unlikely to find optimal living conditions here [14]. This assumption was partially confirmed when in 2008, marked pikeperch were caught 35 days after stocking in the middle section of the reservoir within 7 km from the stocking place, and in 2007, four marked individuals were caught 43 days after stocking in the tributary section, 10 km from the stocking site.

It took about five weeks to detect the marked pikeperch out of the bay in which they had been stocked. The wind direction was variable in the dam section of the Římov Reservoir with the exception of a 20 day period in 2007, when stronger wind blowing from the northeast occurred. This may help 0+ pikeperch in spreading faster and enable some individuals

to reach as far as the tributary section. This theoretical hypothesis of upstream reservoir migration/movement caused by wind is in strong contradiction to the more apparent effect of floods, for example, when all water masses flow in the direction of gravity. In Orlík Reservoir, Čech et al. [3] observed that the floodwaters that swept through the riverine section of this large water body completely shifted the existing pelagic community of early juveniles, especially European perch and pikeperch, from the riverine into the lacustrine section (a 30 km shift of the whole community downstream within 10 days).

In Římov Reservoir, moreover, the original river valley created meanders in which the water currents caused by wind change rapidly. The variation in the water current direction could prevent pelagic larvae from moving from offshore areas to near-shore habitats with high densities, as in large circular and more wind-exposed lakes [2]. In the epipelagic layer of Římov Reservoir, there is no significant surface flow from the Malše River toward the dam that would have to be overcome during upstream migration (due to a lower temperature, the river plunged into the intermediate layer of the water column 0.5–1.5 km from the tributary), and the stocked larvae, with a standard length of about 7 mm, are probably too small to overcome distances in the order of several kilometers. After five weeks, when the marked pikeperch were first captured at considerable distance from the bay, they had a standard length of about 13–15 mm, twice the size at release. Because larger fish have greater swimming ability [39], the fish larvae must remain at the stocking site to grow and transform into juveniles with fully developed fins, so they can overcome larger distances and migrate to more appropriate foraging habitats. After five weeks, marked pikeperch were rarely, but almost exclusively, caught outside the bay in which they were released. One theoretical explanation is the high mortality of early pikeperch stages in the reservoir, and that some stocked fish left their stocking location between the surveys. An unknown part of the 0+ pikeperch can migrate downstream of the reservoir [40]. In 2007 and 2008, nearly half a million pikeperch larvae were stocked into the reservoir. Only 10 days after stocking, during trawling in the bay where the marked pikeperch were stocked, and where therefore the highest density of marked pikeperch should be present, the proportion of marked pikeperch to the total 0+ pikeperch cohort was mostly less than 10% (Table 1). Although this was not the main objective of this study, this fact indicates that the natural reproduction of pikeperch in the reservoir is not negligible. The fact that 0+ fish abundance in late summer in the reservoir is usually very low [41] indicates that larval and early juvenile mortality is high, or that 0+ pikeperch move to areas not sampled (e.g., steep shores with boulders and stumps) or out of the reservoir system.

The diel vertical migrations of percid fish in a deep reservoir were described for only part of the 0+ fish community [6]; the movement activities and vertical migrations of stocked pikeperch might be a behavioral response to predation, food, or phototaxis that is specific to the individual or learned from wild fish, rather than genetically encoded which hasn't been confirmed for European perch either [42]. Since the pikeperch is an extremely valuable species from both an ecological and an economic perspective, it is also extremely important to understand the early stages of their development. Mass marking with oxytetracycline or other fluorescent substances provides an opportunity to study the spatiotemporal dispersal of early pikeperch stages, and could be used in the future for other ecological studies, such as calculating survival rates [43]. A comparison of 0+ pikeperch of both wild and hatchery origin would be an appropriate experiment. As hatched larvae are very sensitive to handling, installation of artificial nests in a natural water body and the removal of eggs at the eyespot stage can solve the problem of obtaining enough wild fish. Given the relatively small number of recaptures (this study), smaller systems should be used for later mark–recapture experiments. It is evident that to use a semi-enclosed bay for such experiments is not fully appropriate, because 0+ fish may disperse out of the bay a few weeks after stocking, and become diluted in the large volume of the reservoir, where their recapture is difficult. Moreover, a more balanced study design with an even greater and more intense sampling effort would help to interpret the movement, survival, and spatial distribution of 0+ pikeperch even more accurately.

## 5. Conclusions

Despite their small body size, the dispersion of 0+ pikeperch can be very rapid, and it is accelerated by water currents; fish stocking at this stage can be carried out in a limited area of a reservoir. Such a fast spreading of early life stages in various directions can have significant implications in case of invasion, as the species could be of significant risk to their new environments. In terms of location within a water body, more productive sites, generally preferred by the species [38], should be selected. A similar conclusion was drawn for stocking juveniles of another piscivorous species, asp (*Leuciscus aspius*), where the greatest dispersal from the stocking site was observed in the least suitable habitat, and in contrast, most of the marked fish were detected in suitable habitats [44]. In a deep water body, 0+ pikeperch can occupy the epipelagic and the bathypelagic layers, which reduces the fish density, potential competition, and predation risk. Therefore, the depth of the stocking waterbody seems to be less important than productivity.

**Supplementary Materials:** The following supporting information can be downloaded at: https://www.mdpi.com/article/10.3390/d15060720/s1.

**Author Contributions:** P.B.: Investigation, Visualization, Writing—Original draft. T.J.: Conceptualization, Investigation, Writing—Review and editing. M.Č.: Investigation, Writing—Review and editing, J.P.: Resources, Supervision, Investigation, Methodology, Writing—Review and editing. All authors have read and agreed to the published version of the manuscript.

**Funding:** The work was supported by the ERDF/ESF project "Biomanipulation as a tool for improving water quality of dam reservoirs" (No. CZ.02.1.01/0.0/0.0/16_025/0007417), project QK23020002 "Pikeperch fry production, their adaptability and optimalization of their stocking into open waters", and CAS within the program of the Strategy AV 21 "Land conservation and restoration".

**Institutional Review Board Statement:** The field sampling was performed in accordance with the guidelines and permission from the Experimental Animal Welfare Commission under the Ministry of Environment of the Czech Republic (ref. no. CZ 01679). The methods and ethics of the study were approved by the Experimental Animal Welfare Commission of the Biology Centre of the Czech Academy of Sciences.

**Informed Consent Statement:** Not applicable.

**Data Availability Statement:** Raw data will be available under reasonable request.

**Acknowledgments:** We wish to thank Jaroslava Frouzová for assistance with laboratory processing, Josef Hejzlar for wind data, Christopher Steer for English proofreading, the four reviewers and the editor for their valuable comments for improving the manuscript.

**Conflicts of Interest:** The authors declare no conflict of interest. The funders had no role in the design of the study; in the collection, analyses, or interpretation of data; in the writing of the manuscript, or in the decision to publish the results.

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
