# Peer review of "High Mobility and Flexibility in the Habitat Use of Early Juvenile Pikeperch (Sander lucioperca) Based on a Mark-Recapture Experiment"

_diversity, doi:10.3390/d15060720_

Round 1
Reviewer 1 Report
Very interesting experimental study with practical application
Author Response
We thank the reviewer for the support of our work. We highly appreciate the positive comment.
Reviewer 2 Report
Dear Blabolil et al (authors):
This was an easily understandable study that was clearly written and expressed. The figure visuals (for example, Figure 1) portray information in an attractive and informational manner. The significance of the study was made clear in the Introduction and the materials and methods were clear. In particular, the discussion was well-balanced and it is further elevated because it includes short-comings and need for improvement in future studies. Clearly this study is valuable for understanding dispersal of early hatched juveniles (in this case pikeperch from hatchery origin). One recomended change is that Table 1 fit into one page, not two. And finally, I see the data was collected (and this study was done) a long time ago. So I commend you for your effort in providing it for publication and review.
Author Response
We thank the reviewer for the detailed reading of our manuscript. Table 1 fits in one page in the current version, thank you for the suggestion. We have to admit, it took a lot of time to finish the manuscript as we were busy with other projects in the meantime. We feel, the data are interesting for the scientific community and therefore we find finally time and the suitable journal to finish the work.
Reviewer 3 Report
This paper shows an interesting description of habitat use and dispersal of early juvenile pikeperch based on mark-recapture data. Considering the importance of restocking programms for the management and conservation of fish species, I think this paper is welcomme. The paper is well written and the results are clearly presented.
Perhaps, Introduction and Discussion should be slightly improved, adding a comparison between the results related to the hatchery-rared juveniles and those previously obtained for wild juveniles. This would be important in the perspective of restocking programms and success.
Author Response
We thank very much for the constructive comment and support of our work. We would like to continue with our work and mark the wild-origin pikeperch, but the pikeperch larvae are very sensitive to manipulation. It will be possible to store artificial nests for pikeperch to spawn and take fertilised eggs at the eyespot stage before hatching. We added this idea in the Discussion section as recommended. So far, we do not know about any study working with pikeperch (Sander lucioperca) wild larvae. Publications with closely relative walleye (Sander vitreum) were added in the original version: Roseman et al. 2005 J Great Lakes Res and Houde et al. 1969 J Fish Res Board Canada.
Reviewer 4 Report
1. How the juveniles of S. lucioperca were identified? Under the microscope? Before thy went to the laboratory, they were manually selected. Which characters were crucial in the identification?
2. Did they have a yolk-sac? A precise photo can show the object of study.
3. In Table 1 the unit of SL is lacking [mm?]
4. Which other species were found in the samples?
5. Do the wild juveniles and recaptured ones differ statistically in SL in the sample analyzed?
6. Is it possible to show graphically (on the diagrams) the range of depths where juveniles were collected (separately for 2007 (night-day) and 2008 (night-day), or between months in two years of study? The depth of occurrence is very important for juveniles that start feeding.
Author Response
We thank the reviewer for his/her time spending with our manuscript and for the constructive comments.
Q1: How the juveniles of S. lucioperca were identified? Under the microscope? Before they went to the laboratory, they were manually selected. Which characters were crucial in the identification?
A1: After each haul, the catch was collected into a clean bucket. Within 20 min of being caught 0+ pikeperch were manually selected by two or three trained researchers. The pikeperch larvae are characterised by relatively large dark spots (chromatophors) randomly distributed on the sides of the body compared to other common species European perch (it has more small spots on the sides of the body in regular rows). All selected fish were placed in bottles uniquely labelled for each haul and frozen until laboratory processing. In the laboratory, the species were once more verified for species identification under Intraco Micro STM 8235410 N stereomicroscope (7–45 magnification) as pikeperch and European perch differ not only by the spots but also by the numbers of post- and pre-anal myomeres. For more details see Pinder (2001).
Pinder, A.C., (2001). Keys to larval and juvenile stages coarse fishes from fresh waters in British Isles. Ambleside: Freshwater Biological Association
Details were added in the Methods section.
Q2: Did they have a yolk-sac? A precise photo can show the object of study.
A2: All pikeperch larvae (during the marking and captured in the reservoir) were free-flowing without yolk-sac. Pictures of pikeperch and European perch made in the laboratory were added to the Supplementary material.
Q3: In Table 1 the unit of SL is lacking [mm?]
A3: We thank for this detailed remark. It is correct, we are sorry for forgetting the units. The unit was added in Table 1.
Q4: Which other species were found in the samples?
A4: The other common species was European perch, which was usually more abundant than pikeperch in the Římov Reservoir. At the end of the sampling period, Cyprinids started to occur in the catch (recognised by small size, needle-shape without evident pigmentation). The most dominant cyprinid species in the reservoir are roach, bleak and bream so the majority of the catch of cyprinid fish was for sure represented by these three species. The identification of cyprinids in this early stage is relatively difficult so for the purpose of this study, in which pikeperch was the target species, cyprinid fish were not identified into species.
More details were added in the Methods section.
Q5:Do the wild juveniles and recaptured ones differ statistically in SL in the sample analyzed?
A5: We thank very much for the question. We made several attempts to statistically analyse the data. The approaches were discussed with colleagues well-skilled in statistical processing. Unfortunately, the data are specific, the numbers of recaptured fish are very low compared to the numbers of wild-origin pikeperch and the statistical tests were weak. The dataset is significantly imbalanced and the statistical comparison of groups with different variability (practically missing) for the recaptured pikeperch is not correct. The mean sizes with variability (standard deviation) are in Table 1. In this table, a reader can see, the sizes (including variability estimates) were very similar and in the case of balanced datasets, the tests will be not significant.
Q6: Is it possible to show graphically (on the diagrams) the range of depths where juveniles were collected (separately for 2007 (night-day) and 2008 (night-day), or between months in two years of study? The depth of occurrence is very important for juveniles that start feeding.
Q6: We thank for the suggestion. We fully agree the habitat is essential for juveniles. In 2007 we focused on the epipelagic layer 0-2 m and in 2008 we dedicate more effort to sampling the bathypelagic layer 7-13 m. The bathypelagic layer was controlled by a scientific echosounder and the position was always under the thermocline, sometimes relatively thin (i.e., 2 m), sometimes wider (i.e., 4 m). The position was driven mainly by the thermocline and the bathymetry. In Figure 1, the bathymetry of the Římov Reservoir (left side) and the positions of sampling sides (right side) are plotted.